# Discovering adaptation-capable biological network structures using control-theoretic approaches

**Priyan Bhattacharya**[1,2,3], **Karthik Raman**[2,3,4]*, **Arun K. Tangirala**[1,2,3]*

**1** Department of Chemical Engineering, Indian Institute of Technology (IIT) Madras, Chennai, India, **2** Robert Bosch Centre for Data Science and Artificial Intelligence (RBCDSAI), IIT Madras, Chennai, India, **3** Initiative for Biological Systems Engineering (IBSE), IIT Madras, Chennai, India, **4** Department of Biotechnology, Bhupat and Jyoti Mehta School of Biosciences, IIT Madras, Chennai, India

* kraman@iitm.ac.in (KR); arunkt@iitm.ac.in (AKT)

**Data Availability Statement:** Codes to reproduce the simulations in this paper are available from https://github.com/RamanLab/SystemsTheoryAdaptation.

## Abstract

Constructing biological networks capable of performing specific biological functionalities has been of sustained interest in synthetic biology. Adaptation is one such ubiquitous functional property, which enables every living organism to sense a change in its surroundings and return to its operating condition prior to the disturbance. In this paper, we present a generic systems theory-driven method for designing adaptive protein networks. First, we translate the necessary qualitative conditions for adaptation to mathematical constraints using the language of systems theory, which we then map back as 'design requirements' for the underlying networks. We go on to prove that a protein network with different input–output nodes (proteins) needs to be at least of third-order in order to provide adaptation. Next, we show that the necessary design principles obtained for a three-node network in adaptation consist of negative feedback or a feed-forward realization. We argue that presence of a particular class of negative feedback or feed-forward realization is necessary for a network of any size to provide adaptation. Further, we claim that the necessary structural conditions derived in this work are the strictest among the ones hitherto existed in the literature. Finally, we prove that the capability of producing adaptation is retained for the admissible motifs even when the output node is connected with a downstream system in a feedback fashion. This result explains how complex biological networks achieve robustness while keeping the core motifs unchanged in the context of a particular functionality. We corroborate our theoretical results with detailed and thorough numerical simulations. Overall, our results present a generic, systematic and robust framework for designing various kinds of biological networks.

## Author summary

Biological systems display a remarkable diversity of functionalities, many of which can be conceived as the response of a large network composed of small interconnecting modules. Unravelling the connection pattern, i.e. design principles, behind important biological

**Funding:** PB acknowledges funding from the Ministry of Education, Government of India. The funders had no role in study design, data collection and analysis, decision to publish, or preparation of the manuscript.

**Competing interests:** The authors have declared that no competing interests exist.

functionalities is one of the most challenging problems in systems biology. One such phenomenon is perfect adaptation, which merits special attention owing to its universal presence ranging from chemotaxis in bacterial cells to calcium homeostasis in mammalian cells. The present work focuses on finding the design principles for perfect adaptation in the presence of a stair-case type disturbance. To this end, the current work proposes a systems-theoretic approach to deduce precise mathematical (hence structural) conditions that comply with the key performance parameters for adaptation. The approach is agnostic to the particularities of the reaction kinetics, underlining the dominant role of the topological structure on the response of the network. Notably, the design principles obtained in this work serve as the most strict necessary structural conditions for a network of any size to provide perfect adaptation.

## 1 Introduction

All living cells display a remarkable array of functions, which can be perceived as the response of a complex, multi-level biological network at a systems level. These complex networks are comprised of a variety of components— biological macro molecules—wired together in exquisite fashion. How the wiring of these components affects system function has been a classic subject of research over the last two decades. A variety of mathematical modeling techniques have been employed to model and predict the function of various biological networks [1–3]. Beyond mathematical modeling, systems theory has been particularly useful to understand and characterize various biological systems [4]. Graph-theoretic tools have also found applications in analyzing and understanding biological networks as functional modules [5–9]. Notably, it has been seen through experiments [10] that the design principles, for any given biological response, are relatively conserved across organisms [11]. For instance, it is well-known that the adaptation (definition to be reviewed shortly) involved in performing bacterial chemotaxis in *E. coli* employs negative feedback. Similarly, an adaptive homeostasis network in higher organisms [12] also uses a negative feedback control strategy [13], suggesting the importance of the role played by the network structure in realizing certain biological functionalities. This observation serves as an essential motivating factor behind the search for minimum networks capable of achieving a given biological functionality.

Besides adaptation, several studies have focused on understanding the emergence of functionalities such as riboswitches, oscillation, toggle switches, and determining the underlying circuitry [14–18], employing methods ranging from brute force searches [18] and rule-based or quantitative modelling [14, 15] to control-theoretic approaches [17]. Tyson J (1975) conceived a two-protein negative feedback model with specific rate kinetics to prove the existence of an invariant Poincaré–Bendixson annulus which can lead to oscillation [15]. Otero-Muras and Banga (2016) conceived of an algorithm based on mixed integer nonlinear programming (MINLP) to deduce the design principles for oscillation along with admissible rate constants [19]. The MINLP approach was later applied to deduce the network structures and admissible parameter regions for adaptation [20] which dealt with a networks with imperfect adaptation *i. e.* large but finite precision and non-zero sensitivity. The results showed that biological networks capable of serving as integral controller are the most suitable candidate for robust adaptation. Li *et al* [16] (2017) employed a brute force search across the topology–parameter space and concluded that incoherent self-loops and negative feedback provide robust oscillation in protein systems. Sontag E and Angeli D (2004) showed the necessity of positive feedback to attain a switch-like behavior which plays a crucial role in cell-fate decision making and

quorum switching [21]. Apart from these, network architectures that aid in the establishment of quality control of glycoprotein based on the respective folding status have also been derived through quantitative modelling of the pathway [22].

Adaptation is defined as the ability of the system output (O) to sense a change in the input (I) from the surrounding environment and revert to its pre-stimulus operating state. From the widely discussed bacterial chemotaxis [13], to the regulation of temperature in a volatile environment, or homeostasis, adaptation is believed to have played a pivotal role in evolution [23]. Typically, adaptation is characterized by two key quantities [11], precision and sensitivity. Precision is the ratio of relative changes of input and output and is quantified as

$$P = \left| \frac{I_2 - I_1}{I_1} \right| \Big/ \left| \frac{O_2 - O_1}{O_1} \right| \tag{1}$$

where, $I_2$ is the new input, $I_1$ is the initial input, $O_2$ is the new output steady-state level, and $O_1$ is the pre-stimulus output level. If $O_2 = O_1$, *i. e.* the system's response returns to *exactly* the pre-stimulus level, the adaptation is known as *perfect adaptation*. On the other hand, sensitivity refers to the ratio to the relative difference between the peak value of the output ($O_{peak}$) and the initial steady-state to that of the input:

$$S = \left| \frac{O_{peak} - O_1}{O_1} \right| \Big/ \left| \frac{|I_2 - I_1|}{I_1} \right| \tag{2}$$

Previously, Ma *et al* [11] (2009) investigated three-protein systems that were capable of adaptation. A three-protein system, including self-loops, involves nine possible interactions, each of which can be positive (activating), or negative (repressing), or absent, resulting in a large number ($3^9 = 19, 683$) of possible network structures or topologies. A brute force study of all the possible structures was carried out assuming Michaelis–Menten kinetics for the protein interactions. Each topology was examined for 10, 000 different sets of parameters leading to over $1.6 \times 10^8$ simulations. The topology–parameter combinations that provided precision and sensitivity more than 10 and 1 respectively were considered capable of adaptation. Their study showed that only 395 topologies could perform robust adaptation. Surprisingly, all of the admissible structures had either negative feedback associated with a buffer species or incoherency in the input node's effects on the output via two different paths. A possible explanation inspired from the analysis of Jacobian matrix of the linearised system was also provided in that work. From this perspective, the condition for perfect adaptation in case of a three node network with different input output node was cast as the minor of the matrix element representing an edge from the output to input node should be zero. Later, other systems such as voltage-gated sodium channels and gene regulatory networks were observed to exhibit adaptation as well. Notably, all the deduced structures employed negative feedbacks [24–26].

Sontag E. (2003) argued from an internal model principle perspective that attainment of adaptation with respect to a step-type disturbance requires an integrator within the system [27]. This, if used for a three-protein network, produces topologies similar to the 395 topologies discussed above. Further, others have suggested specific control strategies like integral feedback to be capable of producing adaptation for a small network (containing three nodes) from an internal model principle and transfer function point of view [28–33]. Rahi *et al* [34] (2017) suggested the supremacy of negative feedback loops over the incoherent feed-forward structures in the context of providing adaptation to periodic responses with varying duration for small scale network structures. We have previously employed a transfer function approach to deduce the design principles for adaptation in a three-node network [31]. The main

arguments were that the condition for perfect adaptation requires the transfer function of the system to be stable and contain a zero on the origin.

Interestingly, the condition of 'zero on the origo' is equivalent to the co-factor condition obtained by Tang and McMillen (2016). Perfect adaptation, as characterised by infinite precision, was argued to be possible if it satisfies the cofactor condition (Necessary condition) along with the attainment of Hurwitz stability (The sufficient condition for stability). These conditions were used to derive the three node admissible topologies that can provide perfect adaptation. Furthermore, the authors argued that given a particular network structure, the cofactor condition along with the stability requirement can provide the condition on the rate constants (parameters) for which the network structure remains adaptive thereby gaining a qualitative understanding about the robustness of that topology [35]. Recently, Araujo and Liotta developed a graph-theoretic method to address the same problem for networks of any size. The condition for perfect adaptation was treated equivalent with the same for infinite precision. Upon analyzing the system matrix of the linearised system, the condition for infinite precision was established as the minor of the matrix element referring to the edge from the output to the input node should be zero. A graph theoretic interpretation of this requirement along with a weaker stability condition rendered presence of either feedback (balancer module) or incoherent feed-forward strategies (opposer module) as the only two ways of providing adaptation for networks with an arbitrary number of nodes and edges. Further, Araujo R. and Liotta L. (2018) conjectured that a balancer module should contain at least one negative feedback in order to retain the stability of the overall network [36]. Wang Y. and Golubitsky M. in 2020, used a similar approach to deduce the necessary design requirements for perfect adaptation in three node networks with different input output configuration [9]. In this case, unlike the previous studies, the disturbance input was considered bounded and Lyapunnov stable (Note: previous consideration on the disturbance input being step type belongs to this class). Further, an extension of the same method to networks of any size revealed that the presence of either feed-forward or loop module or a Haldane structure is necessary for perfect adaptation. Interestingly, the Haldane motifs refer to a biochemical network obtained by considering the three phases (active, inactive and off state) of a single biochemical species where the total concentration at any given point of time is assumed to be constant [37].

The present work provides a generic control-theoretic method using a state-space framework and shows that either negative feedback and incoherent feed-forward loop are necessary conditions for perfect adaptation- a specific scenario of adaptation in general. We for the first time, rigorously prove the conjecture by Araujo *et al* [36] (2018) that a 'balancer module' must contain at least one negative feedback in order to provide stability to the entire network. Further, we argue that there exists a class of negative feedbacks (*i. e.* not all negative feedback can provide adaptation) that can only function as a balancer module admissible for adaptation. In this sense, the necessary conditions obtained are the most stringent among the ones in the literature. Our entire algorithm is independent of the kinetics, barring some minimal assumptions. This approach is in agreement with, and a generalization of the findings from previous studies [5, 11], which have argued that the structure of the network plays a determining role for the governing functionality. The proposed approach enables us to identify all possible control strategies without resorting to a computationally demanding brute-force approach that can achieve perfect adaptation. Unlike previous approaches, we have also taken the non-zero sensitivity in to consideration along with the infinite precision condition while defining the mathematical requirements for perfect adaptation. To this purpose, we have employed the well-known concept of controllability and have been able to explain the reason behind a class of networks such as voltage-gated Sodium channels providing adaptive response for only one step change (the famous toilet-flush phenomenon as described by James Ferell [26]).

Interestingly, the Haldane motifs as described by Wang *et al* [37] (2021) falls in this category if there exists a conservation principle. We argue that a system that meets perfect adaptation is also capable of producing peak response in the minimum time when compared with other non-adaptation response. Further, we propose that the adaptive behaviour is invariant to a canonical downstream connection, which in turn shows the context-independence property of the adaptive networks, as opposed to oscillatory networks [38].

The rest of this article is organized as follows. The Methodology Section presents key concepts leading to the proposed algorithm, where the conditions for perfect adaptation are translated into certain equality constraints on the parameters of systems theory. The question of minimum peak response time is also addressed in this section. In the Application Section, the postulated mathematical conditions are used to identify the potential network structures of any size for adaptation The particular case of retroactivity in adaptation is also explained in the proposed mathematical framework of control theory. The final Discussion Section places the results along with the simulation studies in perspective.

## 2 Methodology

In this section, we outline a generic framework to deduce network structures capable of adaptation. First, we derive the mathematical requirements for the condition of adaptation using linear systems theory. Using these conditions, we first discover the motifs for adaptation by networks with a minimum number of nodes and edges. These conditions are further scaled-up to determine the necessary conditions for adaptation in networks of larger sizes, with arbitrary numbers of nodes and edges.

### 2.1 Linearisation of the rate reactions

Working in the linear domain allows us to utilize the wealth of linear systems theory. Given an enzymatic reaction network, the rate equations for the nodes, *i. e.* enzyme concentrations ($\mathbf{x}$) can be written as

$$\dot{\mathbf{x}}(t) = \mathbf{f}(\mathbf{x}(t), \mathbf{u}(t)), \quad \mathbf{y}(t) = \mathbf{g}(\mathbf{x}(t), \mathbf{u}(t))$$

where $\mathbf{x}(t)$, $\mathbf{u}(t)$ and $\mathbf{y}(t)$ are the states, inputs or known disturbances and output, respectively. For this set-up, the linearized state-space model is

$$\dot{\mathbf{x}}(t) = \mathbf{A}\mathbf{x}(t) + \mathbf{B}\mathbf{u}(t),$$
$$\mathbf{y}(t) = \mathbf{C}\mathbf{x}(t) + \mathbf{D}\mathbf{u}(t)$$

where $\mathbf{A}$, $\mathbf{B}$, $\mathbf{C}$ and $\mathbf{D}$ are obtained as the Jacobians of $\mathbf{f}(\mathbf{x}, \mathbf{u})$ and $\mathbf{g}(\mathbf{x}, \mathbf{u})$ with respect to the $\mathbf{x}$ and $\mathbf{u}$, respectively. The corresponding transfer function can be written as

$$\mathbf{G}(s) = \mathbf{C}(s\mathbf{I} - \mathbf{A})^{-1}\mathbf{B} + \mathbf{D} \tag{4}$$

For the problem under consideration, the output and input are scalar variables. However, the obtained results apply to multiple-input, multiple-output (MIMO) systems. Indeed, a linearized model around a steady state does not always capture the non-linear dynamics accurately. However, since adaptation is a stable (convergent) response, according to the Hartman–Grobman theorem [39], the conditions obtained for adaptation using linear time-invariant (LTI) systems theory serve as sufficient conditions for the same even in non-linear systems.

### 2.2 Conditions for perfect adaptation

Perfect adaptation, as defined above, refers to a system that should be sensitive to changes in the input in its transient phase and be able to drive the response to its previous steady-state

value. These conditions can be translated to restrictions on the state space matrices using LTI systems theory as (i) a non-zero peak value and (ii) a zero final value of the output.

The condition of non-zero peak value translates to a non-zero value of the sensitivity. This condition can be attained by making the output mode of the system *controllable* by the environmental disturbance. This can in turn be guaranteed, if the Kalman controllability matrix, $\Gamma_c$, is full row rank, *i. e.*, for an $N$-dimensional state space with a single input,

$$\text{rank}(\Gamma_c) = \text{rank}\left(\begin{bmatrix} \mathbf{B} & \mathbf{AB} & \cdots & \mathbf{A}^{N-1}\mathbf{B} \end{bmatrix}\right) = N \tag{5}$$

Since the system of rate equations are linearized around a stable fixed point, the initial value of the deviated output (deviation from the stable point) of the linearized system should be zero. These conditions, along with the assumption of linear, exponential stability (matrix $\mathbf{A}$ is Hurwitz), can be mapped onto the parameters of an LTI system for a step-change in the external environment, $u(t)$, as

$$y(t) = \int_0^t \mathbf{C}e^{\mathbf{A}(t-\tau)}\mathbf{B}d\tau + \mathbf{D}u \tag{6}$$

$$y(t=0) = 0 \Rightarrow \mathbf{D} = 0 \tag{7}$$

Using Eq 6, the condition for zero final value can be obtained as

$$y(t) = \mathbf{C}\mathbf{A}^{-1}[e^{\mathbf{A}t} - \mathbf{I}]\mathbf{B} \tag{8}$$

$$\lim_{t\to\infty} y(t) = 0 \Rightarrow \mathbf{C}\mathbf{A}^{-1}\mathbf{B} = 0 \; [\mathbf{A} \text{ is Hurwitz}] \tag{9}$$

It is to be noted that the zero final value condition may not be achieved in several practical scenarios, leading to *imperfect* adaptation [40]. However, we shall limit this discussion to perfect adaptation. In this sense, adaptation and perfect adaptation shall be used interchangeably from here on.

Although Eqs 9 and 5 constitute the main checkpoints for adaptation, several other additional constraints, such as minimum peak time and minimum settling time can play a crucial role in sensing the change in the external disturbance and promptly acting to reject it. We argue below in Theorem thm0 that the peak time for a system is minimum if the condition of zero final value is satisfied:

**Theorem 1**. *For a set $\mathbb{S} \subset \mathbb{D}$ (where $\mathbb{D}$ is the ring of all causal transfer functions with real poles) consisting of stable, minimum phase transfer functions with the same set of poles and differing by a single zero position with each other, the transfer function with zero final value has the minimum peak time.*

*Proof.* To establish this fact, let us assume a proper LTI system $G(s)$ and another system $H(s)$ with same singularities (all real), except a zero at the origin. Assume $y_1(t)$ ($Y_1(s)$), $y_2(t)$ ($Y_2(s)$) and $t_{p_1}$, $t_{p_2}$ to be the step responses and the peak times for $G(s)$ and $H(s)$, respectively.

$$G(s) = K\frac{(s+z_1)\prod_{k=2}^{n}(s+z_k)}{\prod_{i=1}^{m}(s+p_i)}, H(s) = K\frac{s\prod_{k=2}^{n}(s+z_k)}{\prod_{i=1}^{m}(s+p_i)} \tag{10}$$

$$G(s) = H(s) + z_1\frac{H(s)}{s} \tag{11}$$

$$Y_1(s) = Y_2(s) + z_1\frac{Y_2(s)}{s} \tag{12a}$$

$$y_1(t) = y_2(t) + z_1 \int_0^t y_2(\tau) d\tau \tag{12b}$$

$$\dot{y}_1(t) = \dot{y}_2(t) + z_1 y_2(t) \tag{12c}$$

Setting $t = t_{p_2}$,

$$\dot{y}_1(t)|_{t=t_{p_2}} = 0 + z_1 \max(y_2(t)) > 0 \tag{13a}$$

$$t_{p_1} \geq t_{p_2} \tag{13b}$$

The equality in Eq 13b holds only when $G(s) = H(s)$, i.e. $g(t)$ shows perfect adaptation.

It can also be inferred from Fig 1 that the system with one zero at origo takes minimum time to reach at its peak value when disturbed through a step input. The above result can be extended in the case of damped oscillatory systems as well. From (12a), it can be seen that

$$y_1(t) = y_2(t) + z_1 \int_0^t y_2(\tau) d\tau$$

The peak time for $y_2(t)$ is always less than or equal to that of its integral $\int_0^t y_2(\tau) d\tau$ therefore their combination $y_1(t)$ has a peak time always greater than or equal to that of $y_2(t)$. Therefore, Theorem 1 implies that perfect (theoretically infinite) precision also ensures minimum peak time if the positions of the poles and the rest of the zeros are unchanged.

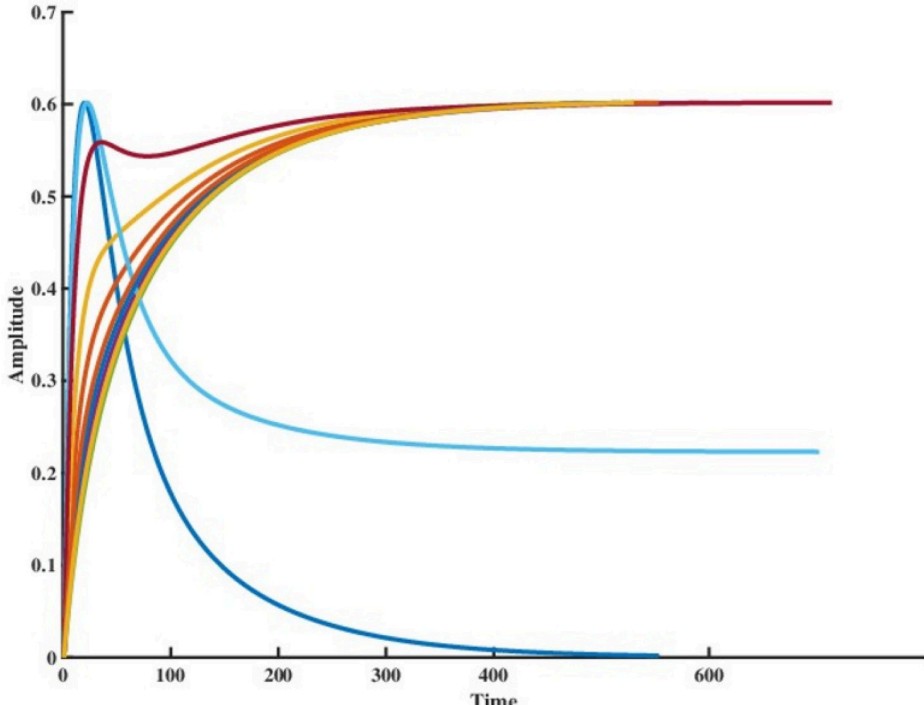

**Fig 1. Comparison of peak times associated with the step responses for a set of ten transfer functions in $\mathbb{S}$ with five poles and four zeros.** It can be seen that the transfer function with zero gain (Response shown in dark blue) provides minimum peak time. The peak values have been kept constant.

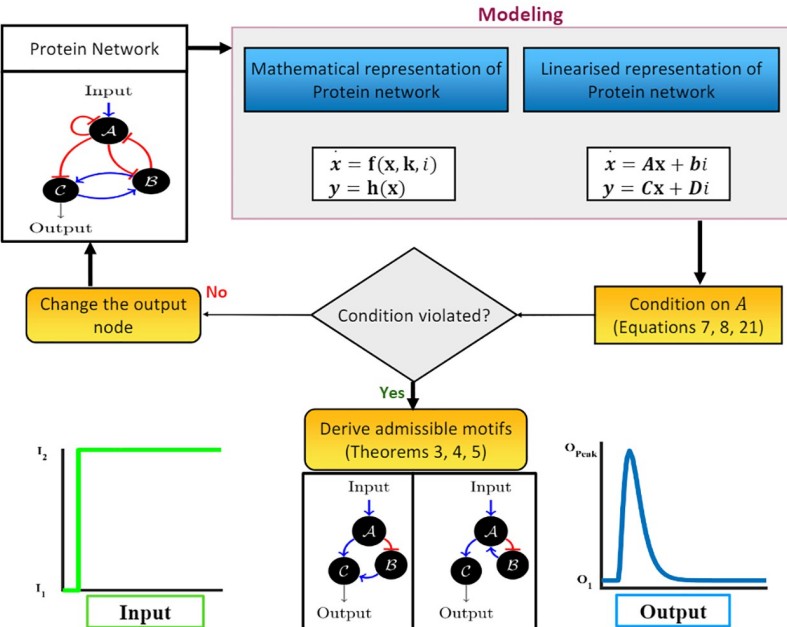

**Fig 2. Workflow of the proposed methodology.** Any given protein network is first linearized, and the conditions on the **A** matrix are investigated, to ultimately derive admissible motifs for the desired functionality.

The minimum settling time requirement involves calculating time constants, which for a large network can be obtained through a simulation study across different sets of time constants while retaining the property of zero final gain ((5)) to ensure perfect adaptation.

To summarize, the conditions for adaptation derived above can be broadly divided into two sets. The first set of conditions (Eq 9) take care of the criteria for infinite precision, which includes the stability of the system matrix **A** and zero final gain of the step input. The second set (Eq 5) ensures non-zero sensitivity. This includes the controllability condition. Moreover, for a given network with a specific input–output configuration (*i. e.* with given **B** and **C** matrix), if the attainment of one set of conditions *ipso facto* violates other, then the network with the given input–output node cannot provide adaptation (see Fig 2). In that case, a modification of the output node (since the input node is fixed for most of the practical cases) may resolve the problem.

## 3 Results

We demonstrate the capability of the methodology we developed above by applying it to protein enzymatic networks, where each node is a protein, and an edge represents either of the following:

1. *Activation*: a protein $\mathcal{A}$ is said to activate $\mathcal{B}$ when $\mathcal{A}$ acts as a transcription factor that binds the active site of the promoter of $\mathcal{B}$ to aggravate the transcription process for the synthesis of $\mathcal{B}$.

2. *Repression*: similarly, if $\mathcal{A}$ acts as a transcription factor to reduce the transcription rate of mRNA, which translates to $\mathcal{B}$.

For a network containing $N$ nodes, there are $3^{n^2}$ numbers of possible network structures. The generalized state equations for an $N$-node network can be written by considering the

normalized concentration of each protein as states:

$$\dot{\mathbf{x}} = \mathbf{f}(\mathbf{x}, \mathbf{k}, d) \tag{14a}$$

$$y = \mathbf{C}\mathbf{x} \tag{14b}$$

where, $\mathbf{x} \in \mathbb{R}^N$ and $\mathbf{k} \in \mathbb{R}^p$ are the states and the parameters associated with the rate equations. In the single disturbance case, $d$ is referred to as the disturbance variable. The protein that receives the external disturbance directly is considered as the input node. The concentration of the $N^{\text{th}}$ node is taken as the output.

### 3.1 Assumptions

It is to be noted that though (14a) does not assume any particular rate kinetics it should satisfy a number of conditions for us to apply our methodology.

1. The vector field ($\mathbf{f}(\mathbf{x})$) in (14a) should be Lipschitz continuous with respect to $\mathbf{x}$ and $d$. This is to ensure the uniqueness of the solution to the dynamical system

2. Given a state $x_i$, the dynamics of the state can be written as

$$\dot{x}_i = f_i = \sum_{j=1}^{N} f_{i,j}(x_i, x_j) \tag{15}$$

   where, $f_{i,j}(x_i, x_j)$ captures the effect of the $j^{\text{th}}$ node on the dynamics of the $i^{\text{th}}$ node.

3. $f_{ij}(\mathbf{x})$ (for activation) or $-f_{ij}(\mathbf{x})$ (for repression) is a class $\mathcal{K}$ (*i. e.* monotone within a finite open interval in the domain, and passes through the origin) function with respect to $x_j$, $\forall j \neq i$.

The above assumptions are not too restrictive, in the sense that almost every form of reaction kinetics prevalent in any biochemical networks satisfy these.

It is important to note that the due to assumption 3, the system matrix $\mathbf{A}$ for an $N$-node system carries not only the necessary information about the structure of the network but also the type of each edge, *i. e.* activation or repression. For instance, if $\mathcal{A}$ represses $\mathcal{C}$, the element in the associated $\mathbf{A}$ matrix that corresponds to this edge turns out to be negative. Intuitively, $\mathbf{A}$ matrix acts as a variant of the incidence matrix for the graphical network, with the diagonals being the exceptions. It is possible to have a negative or non-positive value of the diagonal element, albeit in the presence of a self-activation loop. These inherent properties of the biological systems' rate dynamics perform an instrumental role in maintaining the structural determinism property of adaptation.

### 3.2 Two node networks- are they capable of adaptation?

From a systems theoretic viewpoint, the step response of a first-order system is always a monotone which is not the case with adaptation. Therefore, the possibility of providing adaptation for any single protein can be safely ruled out. The immediate next case of $N = 2$ can be investigated. Implementing the aforementioned approach (Fig 2) reveals that two protein networks with different input and output nodes are unable to provide adaptation. However, as shown in Fig 3, two-node networks can indeed perform perfect adaptation provided the output is measured at the same node that receives the disturbance input. (see section 1 of S1 Text).

Interestingly, there exists a class of biological networks that provide adaptation for a single step input but do not respond to subsequent perturbations [26]. This is defined as the *toilet*

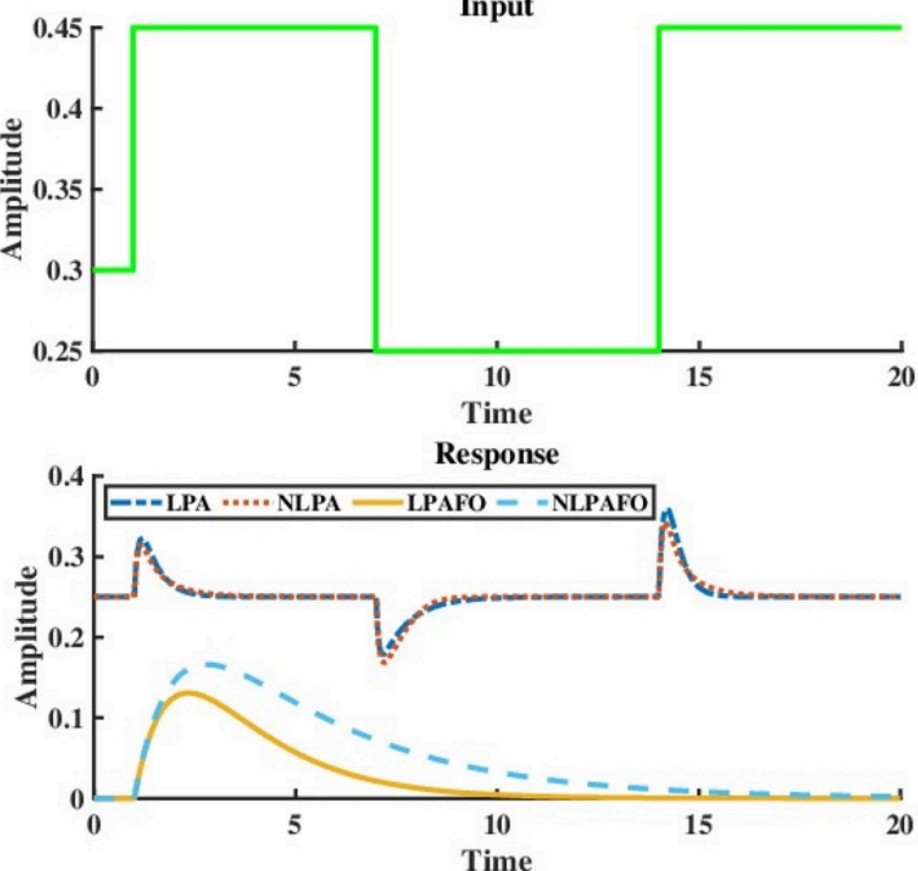

**Fig 3. Two node networks capable of adaptation subject to staircase disturbance.** The abbreviations LPA, NLPA, LPAFO, NLPAFO stand for linear perfect adaptation, non-linear perfect adaptation, linear perfect adaptation for once and non-linear perfect adaptation for once respectively. The network architecture and necessary values for this simulation have been provided in S1 Text.

*flush phenomenon* (Fig 3). Friedlander *et al* [24] (2009) and Goh *et al* [41] (2013) showed that this phenomenon occurs in a three node network with an equality constraint stemming out from a conservation law thereby reducing the effective number of state variables to two. In this regard, the aforementioned algorithm provides a great systems-theoretic perspective to explain and design such networks. If the time difference between two successive step perturbations is large enough (compared to the system's settling time), then the condition for adaptation in this case is the same as that for a single step. Along with this, it is to be observed that with each step perturbation, the steady-state values of the system changes (note that the adaptation property guarantees the invariance of the steady-state of the output state only), which leads to a different linearized model. If the modified linearized model remains controllable and the general condition of adaptation is satisfied, the system provides adaptation for staircase input (Refer to section 1 of the S1 Text file for a detailed discussion).

### 3.3 Three node networks- smallest motifs capable of adaptation?

The admissible network structures obtained from the analysis of the two-node enzymatic networks exclude the possibility of network structures that can provide adaptation with different input-output nodes. Therefore, it is important to identify a control strategy–perhaps the

inclusion of an additional controlling node–that can bring adaptation to the two-node protein system with different input–output nodes.

From the perspective of a control-theoretic framework, the functionality of adaptation can be thought of as a regulation problem. Considering the biological feasibility and the network with only one external disturbance input ($\mathcal{D}$), we propose a feedback control scheme where another protein $\mathcal{B}$ can act as a controller node. If the concentration of $\mathcal{B}$ is $u$, the controller dynamics can be written as

$$\dot{u} = g(x_1, x_2, u) \tag{16}$$

We adopted $g : \mathbb{R}^3 \rightarrow \mathbb{R}$ as a linear function of the states and the control input.

$$g(x_1, x_2, u) = \alpha_{ab}x_1 + \alpha_{cb}x_2 + \alpha_{bb}u \tag{17}$$

The parameters such as $\alpha_{ab}$ and $\alpha_{cb}$ govern the strength and type (repression or activation) of the edges. From feedback control theory [39] if the open-loop system is fully controllable by $u$ then consideration of $u$ as a variant of dynamic state feedback control strategy does not alter the controllability of the system.

**3.3.1 Finding the minimal admissible topologies.**   The closed system can be written as

$$\dot{\mathbf{x}} = \begin{bmatrix} \alpha_{11} & \alpha_{12} \\ \alpha_{21} & \alpha_{22} \end{bmatrix} \mathbf{x} + \begin{bmatrix} \beta_1 & \beta_2 \\ 0 & \beta_3 \end{bmatrix} \begin{bmatrix} d \\ u \end{bmatrix} \tag{18}$$

For the system to provide adaptation, $x_2$ has to be controllable by the control input $u$. For the closed-loop system, the infinite precision condition for adaptation can be written as

$$\exists p \in \mathbb{R} : \left\{ \mathbf{x}^* = \begin{bmatrix} p \\ 0 \end{bmatrix}, u^* \right\} \text{ s.t. } \begin{bmatrix} \dot{\mathbf{x}} \\ \dot{u} \end{bmatrix} = \mathbf{0}|_{\mathbf{x}^*, u^*}$$

For the system with controller,

$$\dot{\mathbf{x}} = \begin{bmatrix} \alpha_{11} & \alpha_{12} \\ \alpha_{21} & \alpha_{22} \end{bmatrix} \mathbf{x} + \begin{bmatrix} \beta_1 & \beta_2 \\ 0 & \beta_3 \end{bmatrix} \begin{bmatrix} d \\ u \end{bmatrix} \tag{19a}$$

$$\dot{u} = \begin{bmatrix} \alpha_{ab} & \alpha_{cb} \end{bmatrix} \mathbf{x} + \alpha_{bb}u \tag{19b}$$

$$\begin{bmatrix} \dot{\mathbf{x}} \\ \dot{u} \end{bmatrix} = \underbrace{\begin{bmatrix} \alpha_{11} & \alpha_{12} & \beta_2 \\ \alpha_{21} & \alpha_{22} & \beta_3 \\ \alpha_{ab} & \alpha_{cb} & \alpha_{bb} \end{bmatrix}}_{\mathbf{A}_{cl}} \begin{bmatrix} \mathbf{x} \\ u \end{bmatrix} + \begin{bmatrix} \beta_1 \\ 0 \\ 0 \end{bmatrix} d \tag{19c}$$

Using the condition for adaptation,

$$\begin{bmatrix} 0 \\ 0 \\ 0 \end{bmatrix} = \begin{bmatrix} \alpha_{11} & \alpha_{12} & \beta_2 \\ \alpha_{21} & \alpha_{22} & \beta_3 \\ \alpha_{ab} & \alpha_{cb} & \alpha_{bb} \end{bmatrix} \begin{bmatrix} p \\ 0 \\ u^* \end{bmatrix} + \begin{bmatrix} \beta_1 \\ 0 \\ 0 \end{bmatrix} d \tag{20}$$

$$\Rightarrow \alpha_{21}\alpha_{bb} - \beta_3\alpha_{ab} = 0 \tag{21}$$

The condition $\alpha_{21}\,\alpha_{bb} - \beta_3\,\alpha_{ab} = 0$ can be achieved in three scenarios:

1.  All the terms are zero: this leads to singularity of $\mathbf{A}_{cl}$, and is hence not acceptable.

2. $\alpha_{21}\,\alpha_{bb} = \beta_3\,\alpha_{ab} = 0$: this is feasible. Interestingly, if $\alpha_{21} = 0$, the state $x_2$ becomes unobservable. Also, in order to attain the condition for adaptation, making $\alpha_{21} = 0$ requires either (i) $\beta_3$ to be zero, which in turn, results making $x_2$ an uncontrollable mode with respect to $u$ or $alpha_{ab} = 0$ leading to uncontrollability with respect to $i$.

3. $\alpha_{21}\,\alpha_{bb} = \beta_3\,\alpha_{ab} \neq 0$: this is acceptable as long as $\mathbf{A}_{cl}$ is Hurwitz.

Combining each of the feasible possibilities with the infinite precision condition for adaptation, we arrive at a superset of admissible motifs from the above possibilities.

As it can be seen from Table 1, the first three network motifs involve negative feedback engaging node $\mathcal{B}$. This type of network can be termed as *negative feedback loop with a buffer node (NFBLB)*. Since NFBLB involves negative feedback, the corresponding response becomes damped oscillatory for most of the cases. However, as long as the adaptation criterion is satisfied, as it can be seen from Fig 4B and 4C, the output after a damped oscillatory transient response goes back to its initial steady state.

The remaining motif carries an incoherency between the two forward paths ($\mathcal{A} \rightarrow \mathcal{C}$ and $\mathcal{A} \rightarrow \mathcal{B} \rightarrow \mathcal{C}$) from $\mathcal{A}$ to $\mathcal{C}$. This is precisely the reason it is called *incoherent feed-forward loop with proportioner node (IFFLP)*. Owing to the structure of IFFLP, the underlying system matrix $\mathbf{A}$ for IFFLP will always have real eigenvalues, thereby eliminating the possibility of oscillatory transients as seen in Fig 4A.

**3.3.2 All possible three-node motifs capable of adaptation.** After finding the minimal network structures—minimal in terms of edges and number of nodes—we extend the above method to find the necessary topological properties, *i. e.* the existence of feedback or feed-forward configurations without any restriction on the number of edges, for the three-node network.

*Remark* 1: For any three-node network, the corresponding system matrix can be written as

$$\begin{bmatrix} \dot{x}_1 \\ \dot{x}_2 \\ \dot{x}_3 \end{bmatrix} = \begin{bmatrix} a_{11} & a_{12} & a_{13} \\ a_{21} & a_{22} & a_{23} \\ a_{31} & a_{32} & a_{33} \end{bmatrix} + \begin{bmatrix} \beta_1 \\ 0 \\ 0 \end{bmatrix} d$$

$$y = \begin{bmatrix} 0 & 0 & 1 \end{bmatrix} \begin{bmatrix} x_1 \\ x_2 \\ x_3 \end{bmatrix}$$

**Table 1. Necessary mathematical conditions for adaptation.** These mathematical conditions can be translated to structural requirements assuming the monotone property of the underlying autonomous dynamical system.

| Possibilities | Final condition |
|---|---|
| $a_{21}\,\alpha_{cb}\,\beta_2 < 0$ | *Gross* −**ve** *feedback.* |
| $\alpha_{cb}\,\beta_3 < 0$ | −**ve** *feedback between* $\mathcal{B}$ *and* $\mathcal{C}$ |
| $\alpha_{ab}\,\beta_2 < 0$ | −**ve** *feedback between* $\mathcal{A}$ *and* $\mathcal{B}$ |
| $\alpha_{21}\frac{\beta_2}{\alpha_{ab}} < 0$ | *Incoherency in* $\mathcal{A} \rightarrow \mathcal{C}$ |

For adaptation, $\begin{vmatrix} a_{21} & a_{22} \\ a_{31} & a_{32} \end{vmatrix} = 0$ and $\mathbf{A}$ has to be Hurwitz.

$$|\mathbf{A}| = \underbrace{\begin{vmatrix} a_{11} & a_{12} & 0 \\ a_{21} & 0 & 0 \\ a_{31} & 0 & a_{33} \end{vmatrix}}_{L_1} + \underbrace{\begin{vmatrix} a_{11} & 0 & 0 \\ 0 & 0 & a_{23} \\ a_{31} & a_{32} & a_{33} \end{vmatrix}}_{L_2}$$

$$+ \underbrace{\begin{vmatrix} a_{11} & a_{12} & 0 \\ 0 & 0 & a_{23} \\ a_{31} & 0 & a_{33} \end{vmatrix}}_{L_3} + \underbrace{\begin{vmatrix} a_{11} & 0 & 0 \\ a_{21} & a_{22} & 0 \\ a_{31} & a_{32} & a_{33} \end{vmatrix}}_{L_4}$$

As it can be seen, the determinant of $\mathbf{A}$ can always be written as a combination of determinants of elementary topologies containing exactly 3 edges. For $\mathbf{A}$ to be Hurwitz, $|\mathbf{A}|$ has to be negative, *i. e.* at least the determinant of any one of these four matrices has to be negative. If any of the first three terms ($L_1$, $L_2$, $L_3$) is negative, it indicates negative feedback. Note the condition $\begin{vmatrix} a_{21} & a_{22} \\ a_{31} & a_{32} \end{vmatrix} = 0$ is 'structurally' satisfied for $L_1, L_2, L_3$ but in the case of $L_4$ it has to be satisfied by the parameters. If $L_4$ is the only negative term, then there exists an incoherent feed-forward loop in the network. Similarly, multiple negative terms represent the presence of both types of motifs. This implies that for any three-node network capable of adaptation with arbitrary edges, the presence of negative feedback or incoherent feed-forward loop is a *necessary condition* (Fig 4D).

Since the negative determinant for $\mathbf{A} \in \mathbb{R}^{3 \times 3}$ is a weaker condition for stability compared to that of $\mathbf{A}$ being Hurwitz, the presence of either or both incoherent feed-forward and negative feedback loops is only a necessary but not sufficient condition for adaptation.

Fig 4 depicts the response of different admissible three-node networks to identical disturbance input. The signals expressed in lines and dots refer to the responses of the non-linear rate dynamics and corresponding linearized counterparts for the corresponding network structure, respectively. For simulation, a variant of Michaelis Menten kinetics is considered. It can be inferred from the figure that the IFFLP always produces hyperbolic responses. The reason behind this can be traced to the spectrum of the underlying system matrix $\mathbf{A} \in \mathbb{R}^{3 \times 3}$ in the linearized dynamics. Due to the absence of any loop in the network, the associated $\mathbf{A}$ matrix for a feedforward network is lower triangular, with the diagonals being the eigenvalues, thereby resulting in hyperbolic responses. Unlike IFFLP, NFBLB can potentially give rise to oscillatory responses along with perfect adaptation.

**Remark 1**. *As established earlier, the infinite precision condition in* Eq 5, *presupposes BIBO stability of the linearized system. From linear systems theory, a linear system is exponentially (asymptotically) stable if and only if the system matrix $\mathbf{A}$ is Hurwitz. The investigation of the Hurwitzness of a matrix of arbitrary size requires computation of all its eigenvalues which is cumbersome for large matrices. We here present a set of necessary conditions to assess the stability of any digraph matrix that is easy to compute and carries important information about the structure of the network.*

*According to the Caley-Hamilton theorem, for any matrix $\mathbf{A} \in \mathbb{R}^{N \times N}$, the characteristic equation can be written as*

$$C_{\mathbf{A}}(s) = Det(s\mathbf{I}_N - \mathbf{A}) = 0 \tag{22}$$

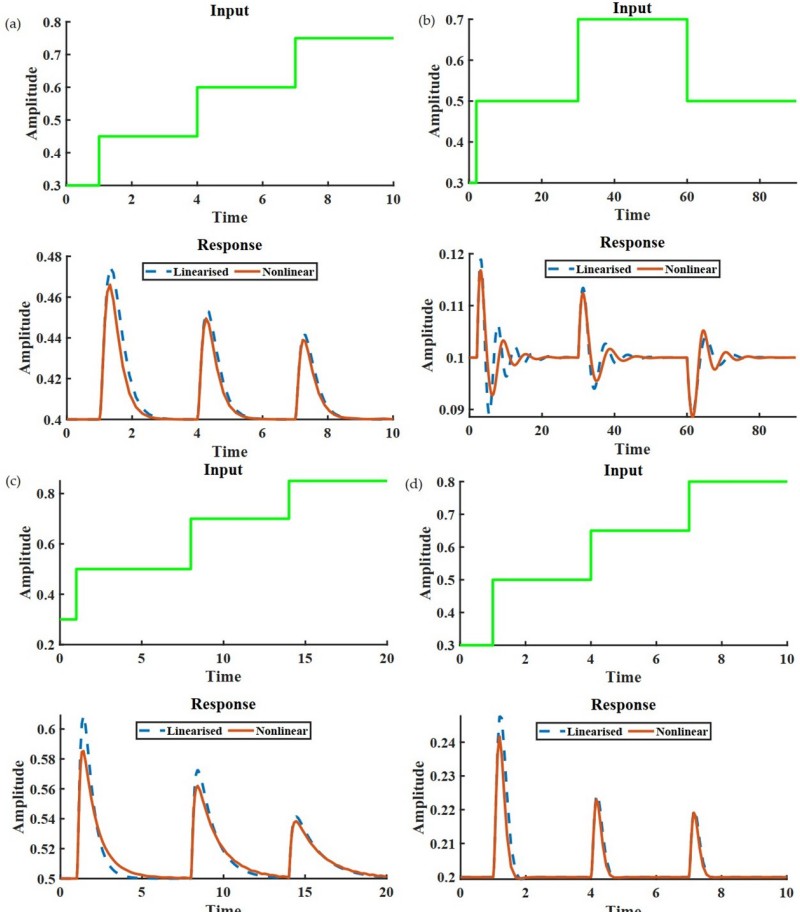

**Fig 4.** (a) shows the response of the output node for a three-node IFFLP topology. (b) shows the same for a three-node NFBLB. The oscillatory behavior can be attributed to the complex eigenvalues of the **A**. Similarly, (c) shows a non-oscillatory response of an NFBLB motif. (d) is the response of the output node of a network containing both the admissible network structure *i. e.* incoherent feedforward path and negative feedback. The network architecture and necessary values for this simulation have been provided in S1 Text.

where, $\mathbf{I}_N$ is the identity element in the field $\mathbb{R}^{N \times N}$ and the roots of these equation are the eigenvalues of $\mathbf{A}$.

Let us denote the roots of Eq 22 as $\lambda_i \in \mathbb{C}, \forall i = 1(i)N$. Therefore, Eq 22 can be written as

$$\prod_{k=1}^{N}(s - \lambda_k) = 0 \tag{23}$$

$$\Rightarrow \sum_{k=0}^{N} C_k s^{(N-k)} = 0 \tag{24}$$

where, $C_k$ is the associated coefficient of $s^{N-k}$ in $C_\mathbf{A}(s)$.

Since, $\mathbf{A}$ is Hurwitz, $Re(\lambda_k) < 0$. Hence, the following set of conditions have to be satisfied for stability

$$C_k > 0, \quad \forall k = 1(i)N \ (C_\mathbf{A}(s) \ is \ monic) \tag{25}$$

*Let us denote the spectrum of $\mathbf{A}$ to be $v := \{\lambda_i\}, \forall i = 1(i)N. C_k$ can be expressed as*

$$C_k = (-1)^k \sum \mathbb{S}_k \qquad (26)$$

*where,*

$$\mathbb{S}_k := \{\sigma_k(v)\} \qquad (27)$$

*where, $\sigma_k$ is the $k^{th}$ order permutation operator that chooses all possible k–tuples of $\lambda_i$'s from $v$ without repetition and the set $\mathbb{S}_k$ stores all such k-tuples. Therefore, the $N^{th}$ condition in Eq 25 is obtained as*

$$C_N > 0 \Rightarrow (-1)^N \sum \mathbb{S}_N > 0$$

*According to the definition of $\mathbb{S}_k$ in Eq 27, there can be only one way to choose the N–tuples of $\lambda_i$'s without repetition from $v$ i. e. the cardinality of $\mathbb{S}_N$ is unity. Therefore,*

$$C_N = (-1)^N \mathbb{S}_N = (-1)^N \prod_{k=1}^{N} \lambda_k > 0 \qquad (28)$$

$$\Rightarrow (-1)^N \prod_{k=1}^{N} |\lambda_k| = (-1)^N Det(\mathbf{A}) > 0 \qquad (29)$$

$$\Rightarrow Sign(Det(\mathbf{A})) = (-1)^N \qquad (30)$$

*Obviously, Eq 30 is a weaker condition for stability than Eq 25 which implies that the necessary structural requirements obtained using Eq 25 are stricter than the ones obtained by Eq 30.*

## 3.4 Structural requirements for adaptation in networks of any size

The above framework, developed for three-node networks, can be extended to larger networks with $N$–nodes and $P$–edges. As shown in the previous section, a three-node network comprising an input, output, and controller can provide adaptation. In this sense, an $N$ ($N \geq 3$) node network can be thought of as the closed-loop system incorporating I/O nodes along with the controller network comprising of the remaining $N − 2$ nodes. In that case, the stability condition for an $N^{th}$ order linear system has to be characterised.

At first, we derive the admissible elementary $N$–node network structures *i.e.* networks that contain at most $N$–edges and can provide perfect adaptation. We then use these results to establish the necessary structural conditions for perfect adaptation in case of any $N$–node network.

**3.4.1 Condition on minimum number of edges in an $N$–node network for adaptation.** In the following theorem, we first derive the lower limit on the number of edges required for an $N$–node network to provide perfect adaptation.

**Theorem 2**. *For a network with $N \geq 3$ nodes, at least N edges are required to provide perfect adaptation.*

*Proof.* It has already been established that in the case of biochemical networks, the system matrix $\mathbf{A}$ for the *linearized* dynamics serves as the digraph generating matrix. Let us assume that the above statement in the theorem is wrong *i.e.* there exists an $N−$ node, $N − 1$ edge network that can achieve adaptation. For an $N$-node, $N − 1$-edge network to show adaptation, it has to satisfy (i) the controllability condition and (ii) infinite precision condition. The mathematical expression for the second has already been derived in Eq 9. However, here we modify the equation for convenience.

$$\dot{\mathbf{x}} = \mathbf{A}\mathbf{x} + \mathbf{B}d \qquad (31)$$

 

where, $\mathbf{x} = \begin{bmatrix} x_1 & x_2 & \cdots & x_N \end{bmatrix}^\tau \in \mathbb{R}^N$ is the state vector with each element ($x_i$) representing the concentration of each node ($i^{th}$ node) and $\mathbf{B} = \begin{bmatrix} \beta & 0 & \cdots \end{bmatrix}^\tau \in \mathbb{R}^{n \times 1}$. Let the output be concentration of the $K^{th}$ node and input be applied on the first node. This implies that the steady state concentration ($x_k^*$) of the output node is zero in linearized representation. At the steady state,

$$\mathbf{A}\mathbf{x}^*_{(x_K^*=0)} + \mathbf{B}d = 0 \tag{32}$$

$$\Rightarrow |\tilde{\mathbf{A}}| = 0 \tag{33}$$

where, $\mathbf{x}^*_{(x_K^*=0)}$ is the steady state solution to Eq 31 with the $K^{th}$ component being zero and $\tilde{\mathbf{A}}$ is the minor of the component in $\mathbf{A}$ representing the edge from the output to the input node.

From elementary network theory, it can be said that it is always possible to design a controllable network of $N$ nodes with $N - 1$ edges if and only if there is no feedback loop. Since the possibility of an isolated node is eliminated, the only feasible structure for a $N$-node, $N - 1$ edge is a feed-forward network ($N$-node networks with a lower number of edges are eliminated for the same reason). Further, since the number of edges is $N - 1$, no node can have more than one incoming or outgoing edge. In order to satisfy the controllability condition, it requires at least one forward path from input node to the output. In the case of an $N$-node network with $N - 1$ edges and no isolated nodes, there can exist one forward path from the input to output node maximum.

The second condition, *i.e.*, the infinite precision condition requires the minor of the component of $\mathbf{A}$ matrix that represents a direct edge from the input to output node be zero. For any digraph matrix in $\mathbb{R}^{N \times N}$, every term in the determinant expression contains $N!$ terms, each a product of $N$–tuples chosen from the matrix. Further, from combinatorial matrix theory [42], each of these $N$–tuples can be expressed as a multiplicative combination of the matrix elements that map to existing loops of the network and the diagonal elements. It can be stated that except the product of all the diagonal elements, every other term in the determinant expression of a digraph matrix maps to the product of the diagonals and the loops. Using this result, it can be claimed that each of the $(N - 1)!$ terms obtained through multiplying $\mathbf{A}_{1k}$ with its minor in the determinant expression of $\mathbf{A}$ is composed of products of the loops and diagonal elements. Also, each of these terms must contain exactly one loop that involves the edge from $k^{th}$ to the first (input) node. According to this result, each term in the minor of $\mathbf{A}_{1K}$ has to contain at least one forward path from the first to the $K^{th}$ node. Since in the case of $N - 1$ edge networks, there can only be one forward path possible, the minor of $\mathbf{A}_{1K}$ is a singleton set. Thus fulfilling the infinite precision condition in this scenario amounts to deleting the only forward path from the input to the output node rendering the system uncontrollable.

On the other hand, it has been already been derived that for a network with $N = 3$, the number of edges required to produce adaptation is also three. By virtue of the foregoing discussion, we conclude that the minimum number of edges required for adaptation is $N$.

**3.4.2 Feedforward networks are adaptive only when incoherent.** We are now ready to present below the most essential and generic results emanating from this work. According to Theorem 2, it requires at least $N$ edges for any $N$–node network to provide adaptation. It can also be shown that there exist only two principal means to satisfy Eq 33 for any elementary $N$–node network (refer to subsection 3.1 of the S1 Text). The admissible elementary network structures can be divided into two further categories i) network without and ii) with loops. In the first scenario, we argue in the following theorem that the existence of at least two opposing feed-forward paths is a necessary condition for adaptation.

 

**Theorem 3**. *For an $N-$node network without any loop, the only way to provide perfect adaptation is to have at least a pair of feed-forward paths from the input to output loop with opposing effects.*

*Proof.* Let us consider the concentration of the $k^{th}$ node as the output variable. It is trivial that in absence of loops, for the output variable to adapt to disturbances, $k$ has to be greater than two. Given an $N-$node, controllable network structure with no loops, it is always possible to order the nodes so that the resultant digraph matrix is lower diagonal. Since the system matrix **A** is equivalent to the digraph matrix, it shall also inherit the lower diagonal structure.

Assuming $k > 2$, for the output node of the network structure to provide adaptation, it has to satisfy the i) controllability (Eq 5) and ii) infinite precision (Eqs 9 and 33) conditions. It can be shown that a feed-forward network is always controllable (refer to Theorem 3 of S1 Text). Also, the lower diagonal property of **A** guarantees the stability of the system, given the diagonals are strictly negative.

The infinite precision condition in Eq 33, requires the minor of the component $\mathbf{A}_{1k}$ (Denote it as $\mathcal{M}$) to be zero. It is important to note here, each of the $(N-1)!$ components of the minor of $\mathbf{A}_{1k}$ is composed of the elements that represent at least one forward path from the input to the output node and the diagonals. Since, there is no loop present in the network the terms in the minor of $\mathbf{A}_{1k}$ should contain exactly one possible forward path from the input node to the output node along with the diagonal elements.

Let us define the set $\mathcal{F}_k \; \forall k = 1(i)N-1$ where each element in $\mathcal{F}_k$ contains the product of the elements in the **A** matrix that represents a forward path with $k$ edges and $N-1-k$ diagonals with no common indices with the former. Consequently, the minor expression can be written as

$$\mathcal{M} = \sum_{p=1}^{N-1}\sum_{j=1}^{N_p}(-1)^p \mathcal{F}_{pj} \tag{34}$$

$$\text{for adaptation, } \mathcal{M} = 0 \tag{35}$$

$$0 = \sum_{p=1}^{N-1}\sum_{j=1}^{N_p}(-1)^p \mathcal{F}_{pj} \tag{36}$$

where, $N_p$ is the cardinality of the set $\mathcal{F}_p$, $\mathcal{F}_{pj}$ is the $j^{th}$ element of $\mathcal{F}_p$. If $\mathcal{F}_{pj}$ has a forward path $f_{pj}$ and the product of the diagonals as $D_{pj}$ the associated cumulative sign $(S_p)$ of $\mathcal{F}_{pj}$ in the minor expression can be written as

$$S_p = (-1)^p sign(F_{pj})sign(D_{pj}) \tag{37}$$

$$\Rightarrow (-1)^p sign(F_{pj})(-1)^{(N-1-p)} \tag{38}$$

$$S_p = (-1)^{N-1} sign(F_{pj}) \tag{39}$$

It is evident from Eq 39, $S_p$ is independent of $p$, the no of nodes engaged in the forward path but a function of the effective sign of the forward path. For Eq 36 to hold, there should be at least one pair with mutually opposed cumulative signs. This can only be possible if there exists at least one forward path with the effective sign being positive, and at least one of the remaining forward paths has to be of the effective sign negative (Fig 5A).

**3.4.3 Conditions on elementary networks with loops for adaptation.** In the second case ($N-$node, $N-edge$ networks with at least one loop), one of the possible network structures with

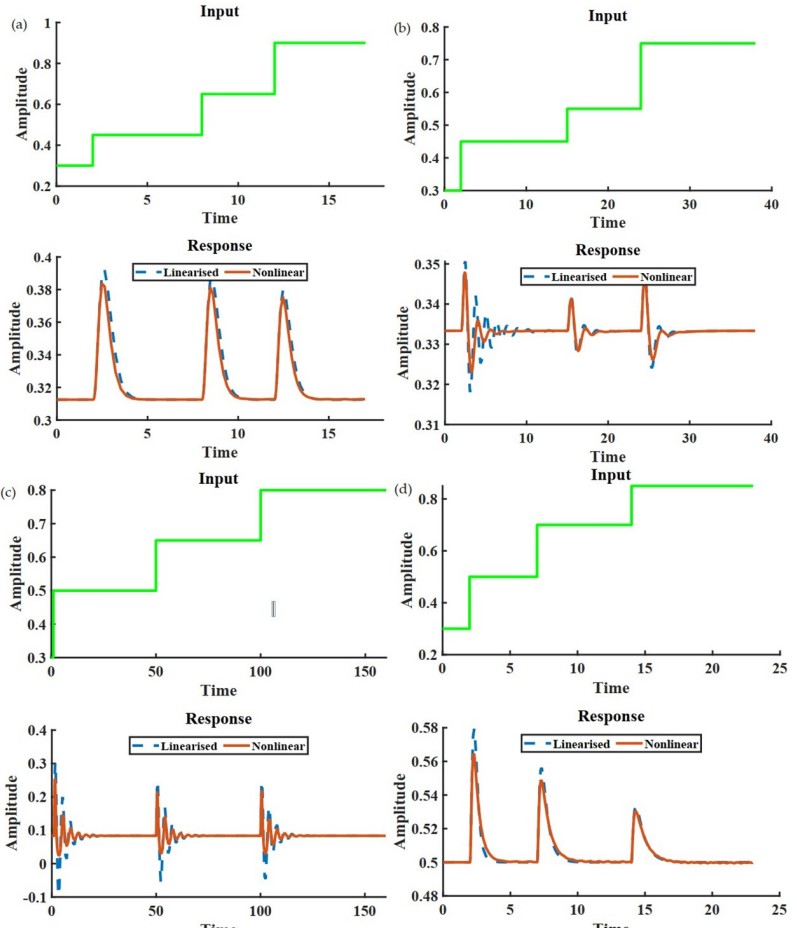

**Fig 5.** (a) shows the response of the output node for a five node IFFLP topology. (b) shows the same for a five node NFBLB with a hyperbolic response. The oscillatory behavior in (c) can be due to negative feedback, leading to complex eigenvalues of the underlying **A** matrix. (d) demonstrates the modular behavior of an NFBLB motif when connected to a downstream system. (e) is the response of the output node of an IFFLP network connected with a downstream system. Although the functionality of adaptation is not compromised, the oscillatory behavior is undoubtedly due to the negative feedback associated with the output of the IFFLP module and the downstream node. The network architecture and necessary values for this simulation have been provided in S1 Text.

$N$ edges can be composed of two or multiple loops without any connecting edge and the common node. In the following theorems, we argue that only a certain class of loop modules can attain adaptation.

**Theorem 4**. *An $N-$ node network containing a single loop with either only one forward path from the input to the output node or multiple coherent forward paths, provides perfect adaptation if the cumulative sign of the feedback loop is negative and it does not contain the edge from output to the input node.*

*Proof.* For the first part of the proof, let us consider an $N-$node network $\mathcal{G}$ containing a single loop $\mathcal{L}_p$ composed of $p$ nodes. Therefore, according to the assumption 3, the corresponding system matrix **A** obtained by evaluating the Jacobian of the nonlinear system at a steady state serves as the digraph matrix for the associated network structure. As it has been shown in remark 1, the necessary condition on $\mathbf{A} \in \mathbb{R}^{N \times N}$ to be Hurwitz can be broken down to $N$ sub conditions. Further, from remark 1 it can be seen that all the $N$ conditions are satisfied for $\mathcal{G}$ if the cumulative sign of $\mathcal{L}_p$ is negative.

For the second part of the proof, let us consider the cumulative sign of $\mathcal{L}_p$ is negative and it also contains the edge from output to the input node. To avoid the trivial case, we assume that there exists at least one forward path from the input to every node in the network. Without any loss of generality, the concentration of the $p$ nodes involved in $\mathcal{L}_p$ are denoted as $\begin{bmatrix} x_1 & x_2 \cdots x_p \end{bmatrix}$ with $x_1$ and $x_k$: $k \in \{2, p\}$ being the concentration of the input and the output nodes respectively. Since there are no more loops in the network, rest of the $N - p$ nodes have to be connected in a feed-forward fashion. Further, there are no edges possible from any of the remaining $N-p$ nodes to the $p$ nodes engaged in loop $\mathcal{L}_p$. We can prove this by contradiction. Let us assume that there exists an edge from $\tilde{p}^{\text{th}}$ node $x_{\tilde{p}}$ (where $\tilde{p} > p$) to $\tau^{\text{th}}$ node $x_\tau$ (where $\tau \leq p$). Since the network is structurally controllable there exists at least one path from the input node to $x_{\tilde{p}}$. Denote it as $\mathcal{S}_f$. Also, since, the input node $x_1$ and $x_\tau$ are in $\mathcal{L}_p$ there exists a path from $x_\tau$ to $x_1$. Denote it as $\mathcal{S}_b$. In this case, if $\mathcal{S}_b$ and $\mathcal{S}_f$ shares a common node $x_c$, then $x_1 \xrightarrow{\mathcal{S}_a} x_c \xrightarrow{\mathcal{S}_f} x_1$ can be conceived as another loop. This violates the assumption of the network containing single loop. Therefore, the resultant structure of the associated $\mathbf{A}$ matrix for the network can be written as

$$\mathbf{A} = \begin{bmatrix} \mathbf{L}_{p \times p} & \mathbf{0}_{p \times N-p} \\ \mathbf{Q}_{N-p \times p} & \mathbf{F}_{N-p \times N-p} \end{bmatrix} \tag{40}$$

It can be inherited from Eq 40 that $\mathbf{F}$ is lower diagonal for it represents a feed forward network structure between the remaining $N - p$ nodes.

For adaptation, the infinite precision condition described in Eq 5 requires the minor of $\mathbf{A}_{1,k}$ has to be zero. Since the output node $x_k$ is involved in $\mathcal{L}_p$, and as shown in section 3.1 in the S1 Text file that for an negative feedback loop to provide perfect adaptation at least one of the diagonal elements in the loop has to be zero, the minor ($\tilde{\mathbf{A}}$) of $\mathbf{A}_{1,k}$ can be written as

$$\tilde{\mathbf{A}} = \mathbf{L}(\mathcal{L}_p)/\mathbf{A}_{1,k} \times \text{Det}(\mathbf{F}) \tag{41}$$

where, $\mathbf{L}(\mathcal{L}_p)$ refers to the term in the determinant expression of $\mathbf{L}$ that represents the loop $\mathcal{L}_p$.

Since $\mathbf{F}$ is lower diagonal, $\text{Det}(\mathbf{F})$ is product of the diagonals which can never be zero for stability purpose which implies the minor can never be zero. Therefore, the network with a single loop and coherent or single forward path from input to the output can never provide perfect adaptation if the loop contains an edge from the output to the input node (Fig 5B).

**3.4.4 General structural requirements for adaptation.** The above important results help us find out the necessary structural conditions for a network of $N$ nodes and $P$ edges, ($\forall N \geq 2$, $\forall P \geq 2$) to provide adaptation that are formalized below.

**Theorem 5**. *Statement 1 serves as a necessary condition for Statement 2*

1. *There exists either at least one negative feedback that does not contain the edge from the output to the input node or incoherent feed-forward loop in the network.*

2. *The network attains perfect adaptation in the presence of a step type disturbance.*

*Proof.* From computational matrix theory it is well known that each term of the determinant expression of the digraph matrix contains a combination of loops and diagonal elements [42]. Further, each term in the coefficient of $s^{N-k}$ in Eq 24 can be of three types i) loops with $k$ edges ii) product of $k$ diagonal elements and iii) possible combinations of $L_k$ number of non-intersecting loops (loops that do not share any common node or edge) totaling $N_k$ number of

edges and product of $k - N_k$ diagonals. Also, in terms of the third type, the total number of edges consumed by a single loop is $k - 2$ given there exists more than one loop in that term.

As we know, for networks with the digraph matrix $\mathbf{A} \in \mathbb{R}^{N \times N}$ to be Hurwitz stable, it has to satisfy Eq 25. Also, any element $E_k$ in the coefficient of $s^{N-k}$ in Eq 24 can be expressed as

$$E_k = (-1)^k \times (S_p) \times L_k \prod D_{k-N_k} \tag{42}$$

where, $S_p \in \{-1, 1\}$ is called the prefix sign attached to each component of $C_k$, $L_k$ refers to the combination of $L_p$ loops with no common nodes engaging $N_k$ edges and $D_{k-N_k}$ are the diagonals concerning the remaining rows of $\mathbf{A}$ matrix.

From combinatorial matrix theory, the prefix sign of each term in the coefficient expression of any matrix can be determined by calculating the minimum number of exchanges needed to arrange them as products of diagonals. Therefore, the sign $S_p$ is determined by the number of co-ordinate exchanges required to perform within $L_k$ ($D_{(k-N_k)}$ is already composed of the diagonals.) such that the combined expression ($\tilde{E}_k$) gets modified to a product of $k$ diagonals. It is easy to verify (refer to the subsection 3.2.1 of the S1 Text file for detailed derivation) that for any loop with $p$ edges, it requires exactly $p - 1$ number of exchanges to modify the loop expression to a product of $p$ diagonals. Similarly, $L_k$ containing $L_p$ loops with a total of $N_k$ edges necessitates $N_k - L_p$ number of exchanges to be transformed to product of $N_k$ diagonals of $\mathbf{A}$ thereby making $S_p = (-1)^{N_k - L_p}$. Again, the sign of $D_{(k-N_k)}$ being composed of $k - N_K$ diagonals can be obtained as $(-1)^{k-N_k}$ given all the diagonals are negative. Therefore the actual sign of $E_k$ can be written as

$$\text{Sign}(E_k) = (-1)^k \times (-1)^{N_k - L_p} \times \text{Sign}(L_k) \times (-1)^{k-N_k} \tag{43}$$

$$\text{Sign}(E_k) = (-1)_p^L \times \text{Sign}(L_k) \tag{44}$$

Since,

$$C_k = \sum E_k \tag{45}$$

$$\Rightarrow \text{Sign}(C_k) = \text{Sign}\left(\sum E_k\right) \tag{46}$$

It can be seen that the element containing the $k$ diagonals correspond to the feedforward structure due to the lower diagonal property of the associated $\mathbf{A}$ matrix (refer to sec 3.1 of the S1 Text file). Also, from Theorem 4, for a network with single forward path (or multiple with all sharing same sign) to provide adaptation, it requires at least one feedback loop that does not contain any edge from the output to the input node. Further, any network with loops to provide perfect adaptation, at least one of the diagonal elements concerning any of the nodes engaged in the loop has to be zero.

From Eq 46, it is clear that if all the loops share common nodes amongst themselves and the network has $N - 1$ diagonal elements then $L_N$ shall contain exactly one loop. In that case, for Eq 25 to be satisfied there has to exist at least one term $E_N$ in the expression of $C_N$ such that

$$\Rightarrow \text{Sign}(E_N) \quad > 0$$
$$\Rightarrow (-1)_p^L \times \text{Sign}(L_N) \quad > 0$$
$$\text{Since } L_p = 1 \Rightarrow \text{Sign}(L_N) \quad < 0$$

This concludes the necessity of the presence of negative feedback loops as admissible elementary motifs in this particular scenario for perfect adaptation. Meanwhile, in the case of digraph

matrices with $<N$ diagonal elements and multiple loops, Eq 44 suggests if any element $E_k$ contains even number of loops each with positive feedback, the accumulated sign shall be positive. We argue here that if all the loops of the networks are positive, then it can not provide perfect adaptation.

Consider a structurally controllable network with no incoherent feed-forward characteristics and the $j^{\text{th}}$ diagonal element of the corresponding digraph matrix $\mathbf{A}$ is zero. The co-efficient expression of $s^{N-k}$ in the characteristics equation of $\mathbf{A}$ contains $^{(N-1)}C_k$ numbers of elements each of which is made out of $k$ diagonal components. Let us define a set $\mathbb{S}^{K-p}$ containing determinants obtained from choosing all possible $K-p$ columns from matrix $\mathbf{A}$. The set $\mathcal{L}_j^p$ contains all the $p$-node loops involving node $j$. Therefore, the expression of $C_k$ can be written as

$$C_k = \sum_{p=2}^{k} \sum \mathcal{L}_{i,j}^p \left( (-1)^{k+p-1} \mathbb{S}_{\mathcal{L}_{i,j}^p}^{(k-p)} \right) + R_k + \sum_{i=1}^{N_{C_k}} (-1)^k \sigma_k^i \mathrm{Diag}(\mathbf{A}) \ \left( \mathbb{S}_{\mathcal{L}_{i,j}^p}^{(k-p)} := \sum \left[ \mathbb{S}^{k-p} / \mathbb{D}_{\mathcal{L}_j^{k-p}} \right] \right) \quad (47)$$

where, $\mathbb{D}_{\mathcal{L}_j}^{k-p} \subset \mathbb{S}^{k-p}$ contains determinants of all possible $k-p$ columns and rows of $\mathbb{A}$ except the $j^{\text{th}}$ row (and ipso facto the $j^{\text{th}}$ column), $R_k$ contains the remaining elements made up of loops other than the elements of $\mathcal{L}_j^p$ and the diagonals, and $\sigma_k(\mathrm{Diag}(\mathbf{A}))$ refers to the set of all possible $k$ diagonals chosen from $\mathbf{A}$ without repetition.

We argue that, apart from the third term of Eq 47, the first and second terms individually are negative if all the existing loops are of positive sign. As it can be seen this assertion holds true for $j = 1(i)3$, (refer to subsection 3.1.1 of S1 Text file). We here assume that there exists a non-zero finite integer $k-1$ up to which the above assertion holds true. As shown earlier, for elements in $C_k$ with single loop, the sign of the term is negative if the signs of the underlying loop is positive.

Therefore, for $C_k$ to be positive either the of the two term of Eq 47 has to positive. Let us first consider the first term is positive. Let the cardinality of the set $\mathcal{L}_{i,j}^p$ be $Q(p)$. Thus, given $p$, for the first term to be opposite at least one of $Q(p)$ elements (denote it as $Q_p$) in $\sum \mathcal{L}_{i,j}^p (-1^{k+p-1} \mathbb{S}_{\mathcal{L}_{i,j}^p}^{(k-p)})$ has to be positive. Since, we are only considering positive feedback all the elements of $\mathcal{L}^p$ is positive. Therefore, $(-\mathbb{S}_{\mathcal{L}_{Q_p,j}^p}^{(k-p)})$ has to be positive and greater than other elements in absolute value which, in turn, is possible if at least one of the elements of $-\mathbb{S}_{\mathcal{L}_{Q,j}^p}^{(k-p)}$ (Denote it as $t^{\text{th}}$ element) is positive and greater than others. Assume it is the $t^{\text{th}}$ element of $-\mathbb{S}_{\mathcal{L}_{Q,j}^p}^{(k-p)}$. It is to note at this juncture that according to the assumption for $i < k$ the sign of both the first and second term are negative. Since $p \geq 2 \Rightarrow k-p+1 < k$ therefore, only the third term $i.\,e.$ sum of product of $k-p+1$ diagonals are positive. If $\left( -\mathbb{S}_{\mathcal{L}_{Q_p,j}^p}^{(k-p)} \right)$ is positive then the second and the third term of $E_{k-p+1}$ can be combined as

$$R_{k-p+1} + \sum_{i=1}^{N_{C_{k-p+1}}} (-1)^{k-p+1} \sigma_{k-p+1}^i \mathrm{Diag}(\mathbf{A}) = \sum \left( (-1)^{k-p+1} \mathbb{S}_{\mathcal{L}_{Q_p,j,l}^p}^{(k-p)} \alpha_l \right) \quad (48)$$

Through a constructive proof provided in subsection 3.2.2 of the S1 Text file, it can be shown that $\alpha_t$ is the largest of all $\alpha_l$. Also, since from the condition derived for the first term of $C_k$, $\left( (-1)^{k-p+1} \mathbb{S}_{\mathcal{L}_{Q_p,j,t}^p}^{(k-p)} \right)$ is also the greatest in the set $\left( (-1)^{k-p+1} \mathbb{S}_{\mathcal{L}_{Q_p,j}^p}^{(k-p)} \right)$. Therefore, the term

$R_{k-p+1} + \sum_{i=1}^{N_{C_{k-p+1}}} \sigma_{k-p+1}^i \mathrm{Diag}(\mathbf{A})$ is negative rendering overall $E_{k-p+1}$ to be of negative sign (as

per the hypothesis the first term of $E_i$ is always negative for $i < k$). This violates the stability criterion shown in Eq 25.

For the second possibility, the second term, $R_k$ of Eq 47 is made up of all the loops without the $j^{th}$ node along with the diagonal elements of $\mathbf{A}$. We first set aside the terms containing diagonal elements and only one loop with more than $m$ edges where $m = \lfloor \frac{k}{2} \rfloor$. We define this as $\tilde{R}_k$. Since, there exists only one loop and all the loops considered are of the positive sign, all the elements of $\tilde{R}_k$ are negative. From this logic, the remaining elements in $R_k$ should contain at least one loop with less than or equal to $m$ edges. Again, if $\mathcal{L}^p$ be the set of all the loops with $p$ edges in the network, then $\mathcal{L}^p/\mathcal{L}_j^p$ refer to all the $p-$edge loops that do not engage node $j$.

$$R_k = \sum_{p=2}^{m} \sum_i (\mathcal{L}^p/\mathcal{L}_j^p)_i \left( (-1)^{k+p-1} \mathbb{S}_{\mathcal{L}_{i,(\mathcal{L}^p/\mathcal{L}_j^p)_i}^p}^{(k-p)} \right) + \tilde{R}_k \tag{49}$$

For $R_k$ to be positive at least one of the terms in $\sum_{p=2}^{m} \sum_i \left( \mathcal{L}^p/\mathcal{L}_j^p \right)_i (-1)^{k+p-1} \mathbb{S}_{\mathcal{L}_{i,j}^p}^{(k-p)}$ has to be positive. That is possible if at least one of the product of $p-$edge loop without the node $j$ and the corresponding term in $(-1)^{k+p-1} \mathbb{S}_{\mathcal{L}_{i,j}^p}^{(k-p)}$ is positive. Let us denote for a given $p$, the $w^{th}$ term is positive. Then

$$R_k = \sum_{p=2}^{m} \sum_{i \neq w} (\mathcal{L}^p/\mathcal{L}_j^p)_i \left( (-1)^{k+p-1} \mathbb{S}_{\mathcal{L}_{i,(\mathcal{L}^p/\mathcal{L}_j^p)_i}^p}^{(k-p)} \right) + O \tag{50}$$

$$\text{where, } O = \left( (\mathcal{L}^p/\mathcal{L}_j^p)_w - \underbrace{\prod \mathbf{A}_{i,i}}_{i \in \text{node}(\mathcal{L}^p/\mathcal{L}_j^w)_w} \right) \left( (-1)^{k+p-1} \mathbb{S}_{\mathcal{L}_{i,j}^p}^{(k-p)} \right) + \tilde{R}_k \tag{51}$$

It is already established in the first case that, $\sum_i \left( \mathcal{L}^p/\mathcal{L}_j^p \right)_i$ is always negative given all the loops in the network are positive. Further, it is possible to write

$$\sum_l \left( (-1)^{k+p-1} \mathbb{S}_{\mathcal{L}_{l,j}^p}^{(k-p)} \frac{\alpha_l}{\max(\alpha)} \right) > 0 \tag{52}$$

For $R_k$ to be positive,

$$\left( (\mathcal{L}^p/\mathcal{L}_j^p)_w - \underbrace{\prod \mathbf{A}_{i,i}}_{i \in \text{node}(\mathcal{L}^p/\mathcal{L}_j^w)_w} \right) > \sum_{i \neq w} (\mathcal{L}^p/\mathcal{L}_j^p)_i \left( (-1)^{k+p-1} \mathbb{S}_{\mathcal{L}_{i,(\mathcal{L}^p/\mathcal{L}_j^p)_i}^p}^{(k-p)} \right)$$

Again, if this holds true it can be shown via proof by construction that $E_{p+1}$ is negative rendering the matrix $\mathbf{A}$ unstable. Therefore, it can be concluded that in Eq 47, both the first and second term has to be individually negative in order to satisfy all the $N$ stability conditions shown in Eq 25.

Since, there are $N-1$ diagonal elements, every element in $E_N$ consists exactly one loop concerning the $j^{th}$ node. Therefore, the elements of $E_N$ can be grouped as following

$$C_N = \sum_{p=2}^{N} \sum \mathcal{L}_{i,j}^p \left( (-1)^{N-p+1} \mathbb{S}_{\mathcal{L}_{i,j}^p}^{(N-p)} \right) \tag{53}$$

Now, it has already been proven that for any $k \leq N$, the coefficients $E_1, \cdots, E_{k-1}$ have to be individually positive

$$\sum_{p=2}^{N} \sum \mathcal{L}_{i,j}^{p}\left((-1)^{N-p+1}\mathbb{S}_{\mathcal{L}_{i,j}^{p}}^{(N-p)}\right) < 0 \tag{54}$$

This implies that for $C_1, \cdots, C_{N-1}$ to be positive, $C_N$ has to be negative. This implies violation of the stability criterion in Eq 25. Therefore, given all the loops of a network are of positive sign and at least one diagonal component of the corresponding $\mathbf{A}$ matrix is zero, the system becomes unstable thereby failing to provide adaptation.

It is to be stressed that these structural conditions for adaptation only serve as necessary conditions for two reasons. Firstly, the sign of the determinant condition used here in Eq 25 is only a weak (necessary) property of a stable system. Secondly, there are additional quantitative constraints that are to be satisfied by fine-tuning the parameters. For instance, in a three-protein system, the negative feedback requires $\alpha_{bb} = 0$, which needs to be guaranteed by the parameters. Similarly, a three-protein network with incoherent feed-forward loop requires $\alpha_{21}$ $\alpha_{bb} = \beta_3 \alpha_{ab} \neq 0$ to be satisfied by the parameters.

Interestingly, it is found that adaptation is preserved against the connection with a downstream system (Fig 5C and 5D). The connection considered here is canonical, *i. e.* only the output node is connected with the downstream network.

**Lemma 6**. *If the stability of the system is not altered, then the functionality of perfect adaptation for an upstream system does not get altered if the output node is connected with a downstream system.*

*Proof.* Given an upstream adaptive network containing $N$ nodes and $P$ edges, it is to be proved that the system preserves its functionality if it is connected with another arbitrarily connected network. Without any loss of generality, let us assume the 1st and the $N$th nodes are the input and output nodes of the upstream network, respectively. The downstream system is connected in a feedback fashion with the output node.

Let the system matrices of the upstream and downstream networks be $\mathbf{A}_1 \in \mathbb{R}^{N \times N}$ and $\mathbf{A}_2 \in \mathbb{R}^{P \times P}$, respectively. As per the statement, the upstream system can provide adaptation, *i. e.* $det|\tilde{\mathbf{A}}_1| = 0$, where $\tilde{\mathbf{A}}_1$ is the matrix associated with the minor of $a_{1N}$. Due to the assumption of the structure, the modified system matrix $\mathbf{A}'$ for the augmented system can be written as

$$\mathbf{A}' = \begin{bmatrix} \mathbf{A}_1 & \mathbf{E}_1 \\ \mathbf{E} & \mathbf{A}_2 \end{bmatrix}$$

where, the elements of $\mathbf{E}_1 \in \mathbb{R}^{N \times P}$ are zero everywhere other than the $N$th row. Similarly, the elements of $\mathbf{E} \in \mathbb{R}^{P \times N}$ are zero everywhere other than the $N$th column. For the combined system to produce adaptation, the minor of $a'_{1N}$ has to be zero. The matrix associated with the minor of $a'_{1N}$ ($\tilde{\mathbf{A}}_1'$) can be written as

$$\tilde{\mathbf{A}}_1' = \begin{bmatrix} \tilde{\mathbf{A}}_{1(N-1 \times N-1)} & \mathbf{E}_{2(N-1 \times P)} \\ \mathbf{0}_{(P \times N-1)} & \mathbf{A}_{2(N-1 \times N-1)} \end{bmatrix}$$

Since $\tilde{\mathbf{A}}_1'$ can be expressed as a block diagonal matrix with the lower non-square matrix being zero, the determinant is the product of the individual determinants of $\tilde{\mathbf{A}}_1$ and $\mathbf{A}_2$. According to the assumption on the upstream system $det(\tilde{\mathbf{A}}_1) = 0$, therefore the matrix $\tilde{\mathbf{A}}_1'$ is singular.

This, in turn, implies that the combined system can provide adaptation if the stability is not altered.

This is intuitively a well-expected result because, typically, adaptation networks are mounted on the big downstream network to provide robustness with respect to external disturbances, and the above lemma shows that the adaptation networks are not retroactive and context-dependent.

## 4 Discussion

Biological networks are complex yet well-coordinated and robust in nature. Although the form of the reaction dynamics underlying a network governs specific characteristics of the biological system, the major roles of controlling and coordinating different levels of hierarchy in the networks can be attributed to the very structure of the network. Previous research works have adopted one of a brute-force, graph-theoretic or a rule-based approach for identifying admissible structures for perfect adaptation. The nature of results obtained from these approaches are limited by the computational cost, inability to capture all necessary structures and/or the challenges in handling networks of arbitrary sizes. In this work, we appeal to the linear systems and control theory for obtaining formal and generalised results without being bounded by any of the aforementioned limitations.

Intuitively, it is apparent that for any (biological) system to exhibit adaptation, it should internally possess a feedback and / or feedforward configuration as mandated by control theory. However, deeper and concrete answers, especially on how such results scale-up with the size of network, inevitably call for a formal study. The primary questions that formed the basis of this work are (for perfect adaptation) (i) how do these intuitions formally manifest in biological networks? (ii) what are the possible signature structures and very importantly (iii) whether a generalised result can be obtained for networks of any size? These are somewhat formidable questions, especially given the non-linear nature of biological processes. However, it turns out that linear systems theory can still provide concrete answers. Essentially, the linearized structure of the system provides the answer to a binary question of whether the network is able to provide adaptation or not. If yes, further conditions on the linearized system are obtained and the problem of determining suitable network structure is resolved. The proposed framework is systematic and generic as against computationally demanding search methods and finding specific control strategies for a particular network to achieve adaptation.

Deriving the necessary conditions for adaptation, we show that a minimum of $N$ edges are required for an $N$-node network to produce adaptation. We use this result to deduce further, that there exist only two ways, namely (1) feedback loop, and (2) multiple forward paths in an $N$−node network, to provide adaptation.

Finally capturing the above results in Theorem 5, we show that existence of either a negative feedback loop or incoherent feed forward node acts as a necessary condition for adaptation. This result agrees with the observations in the seminal work of Ma *et al* [11] (2009), in their seminal study of three node networks. Furthermore, this also proves the conjectural assertion by Araujo R. *et al* [36] (2018) that in absence of an opposer module a balancer requires at least one negative feedback to provide perfect adaptation. Lemma 6 establishes that adaptation is retained in presence of a canonical downstream connection. This non-retroactive nature of these networks implies that they are highly likely to preserve their function in synthetic circuits designed with various modules.

It should also be noted that the topologies obtained from the linearized hyperbolic system provide perfect adaptation in the practical (nonlinear) scenario. The more generic case comprising of the possibility of a non-hyperbolic system providing adaptation can be an interesting

future study. Also, the controllability condition used in this paper works as a sufficient condition for the controllability of the actual nonlinear system. The area of nonlinear controllability can be explored in this context to avoid missing out on false negatives.

In sum, we see four definitive contributions of this study. We first proved via Theorem 1 that the network structures for adaptation ipso facto reduce peak time because of the infinite precision (zero-gain) requirement. Secondly, the control-theoretic approach enabled us to address the question of non-zero sensitivity for the first time along with the standard infinite precision requirement for perfect adaptation. In this context, concise conditions inspired from systems theory were proposed regarding the well-known toilet-flush phenomena for the first time. Third, we argue that the structural conditions obtained as the necessary conditions for adaptation herein, are most stringent among the ones in the existing literature (Refer to S1 Table). We make this claim on the basis of two results obtained in this work. Firstly, in Theorem 5, we proved that the sign of at least one feed back loop has to be negative for ensuring adaptation in absence of opposing forward paths. This is a significant reduction for it eliminates the possibility of balancer module with no negative feedback providing adaptation. Secondly, we showed in Theorem 4 that negative feedback loops that do not contain any edge from the output to the input node are the only modules admissible for perfect adaptation in absence of incoherency of the forward paths. Fourth and most notably, the entire algorithm remains agnostic to the particularities of the reaction kinetics. Our approach lays the foundation for the application of LTI systems theory to predict topologies and fine-grained constraints for robust adaptation. Further, as an extension to this approach, building a nonlinear dynamical systems-theoretic approach to unravel the design principles of adaptation can be a promising area of future study.

## Supporting information

**S1 Text. Necessary proofs and illustrations for the main text.**
(PDF)

**S1 Fig. Two-node network.**
(EPS)

**S2 Fig. Proposed five-node network.**
(EPS)

**S3 Fig. Two-node network for simulation.**
(EPS)

**S4 Fig. Voltage gated channels used for simulation.**
(EPS)

**S5 Fig. Three-node IFFLP used for simulation.**
(EPS)

**S6 Fig. Three-node NFBLB used for simulation.**
(EPS)

**S7 Fig. Three-node IFFLP+NF used for simulation.**
(EPS)

**S8 Fig. Five-node IFFLP used for simulation.**
(EPS)

**S9 Fig. Five-node NFBLB used for simulation.**
(EPS)

**S10 Fig. Three-node IFFLP along with a downstream node.**
(EPS)

**S11 Fig. Three-node NFBLB along with a downstream node.**
(EPS)

**S1 Table. Consolidated table: demonstration of the algorithm.**
(PDF)

## Acknowledgments

The authors thank K R Ghusinga for valuable comments on the manuscript.

## Author Contributions

**Conceptualization:** Karthik Raman, Arun K. Tangirala.

**Formal analysis:** Priyan Bhattacharya.

**Funding acquisition:** Karthik Raman, Arun K. Tangirala.

**Investigation:** Priyan Bhattacharya.

**Methodology:** Priyan Bhattacharya, Karthik Raman, Arun K. Tangirala.

**Supervision:** Karthik Raman, Arun K. Tangirala.

**Validation:** Priyan Bhattacharya.

**Writing – original draft:** Priyan Bhattacharya.

**Writing – review & editing:** Priyan Bhattacharya, Karthik Raman, Arun K. Tangirala.

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
