## [Decision Letter · Decision Letter 0]

16 Sep 2021

Dear Prof Tangirala,

Thank you very much for submitting your manuscript "Discovering adaptation-capable biological network structures using control-theoretic approaches" for consideration at PLOS Computational Biology.

As with all papers reviewed by the journal, your manuscript was reviewed by members of the editorial board and by several independent reviewers. In light of the reviews (below this email), we would like to invite the resubmission of a significantly-revised version that takes into account the reviewers' comments.

We cannot make any decision about publication until we have seen the revised manuscript and your response to the reviewers' comments. Your revised manuscript is also likely to be sent to reviewers for further evaluation.  Note that the reviewers raised a major concern that many of the conclusions in your manuscript are not new. In order for the journal to accept your revised manuscript we will need to assess the level of novelty in your results, so it is crucial that you address this major concern very clearly.

Sincerely,

Pedro Mendes, PhD

Associate Editor

PLOS Computational Biology

Jason Haugh

Deputy Editor

PLOS Computational Biology

Reviewer's Responses to Questions

**Comments to the Authors:**

Reviewer #1: In this manuscript, the authors have studied the topic of discovering adaptation-capable biological networks. This has general interests in several research fields. The authors have simplified the problem to studying linear time-invariant systems. And they assume explicitly that one of the nodes be a controller for the adaptation system. Several conclusions interest me. For example, in section 2, the authors talked about how to get a non-zero response; and on page 13, it is neat to rewrite the determinant A into the addition of different components. But apart from these minor points, the current manuscript does not have enough progress to warrant a publication.

The primary focus of the manuscript is to derive the adaptation network’s topological features with control theory. But I feel that the approach is not fundamentally different than studying the Jacobian matrix of ODE equations. And the conclusions are largely the same as the previous findings.

The authors have stated several conclusions in the manuscript. Based on the abstract, the authors have:

1. The authors “translate the necessary qualitative conditions for adaptation to mathematical constraints using the language of systems theory, which we then map back as ‘design requirements’ for the underlying networks. We go on to prove that a protein network with different input–output nodes (proteins) needs to be at least of third-order in order to provide adaptation.”

2. The authors “show that the necessary design principles obtained for a three-node network in adaptation consist of negative feedback or a feed-forward realization. Interestingly, the design principles obtained by the proposed method remain the same for a network of arbitrary size and connectivity.”

3. The authors “prove that the motifs discovered for adaptation are non-retroactive for a canonical downstream connection. This result explains how complex biological networks achieve robustness while keeping the core motifs unchanged in the context of a particular functionality.”

Points 1 and 2 are not new and have been demonstrated in different papers. The third part is not clear.

The manuscript did not emphasize enough what’s new in their study. What I find interesting is 1) the discussion on the network's response, which is not well-studied based on my knowledge. 2) to apply the method to higher-order networks. This topic has been studies but it is still very interesting if the authors could summarize new principles based on their own work. The authors have mentioned both parts, but the conclusions are not clear enough to guide further study for either thermotical or experimental biologists. And it’s not clear from the writing whether the authors need to assume a single control node for large networks.

The writing of the manuscript needs to be improved. First, as mentioned before, the manuscript is not focused enough. Second, the use of figure space is poor. For example, figures 3, 4, and 5 look similar to what’s shown in the figure, yet only after reading the figure legends I realized that they are trying to tell entirely different stories. And many times, I found the legends do not have a graphic counterpart in the figure. Third, some of the contents, like theorem 4, looks obvious to me. If there are no links between the two parts of the networks, it is hard to imagine how they will influence each other. Maybe the rigorousness of mathematical requires contents like this to be presented. I don’t see the necessity from a practitioner’s point of view.

Reviewer #2: The manuscript is clearly-written and describes a timely subject that is of interest in the field of understanding and designing biological feedback mechanisms.

However, I do have a major concern: I am not clear on how the work distinguishes itself from several previously-published papers. I hope that the authors can address the ways in which the current work differentiates itself from the key results reported in the following:

Ref 7 - Wang et al. (2016) - This is referenced in passing in the Introduction, as a member of the group of graph-theory approaches to the problem, but it seems that the results in that paper have considerable overlap with the ones discussed in the Results section, including the nature of topological constraints on networks with various numbers of nodes.

Ref 11 - Ma et al. (2011) - This is discussed extensively in the current manuscript, but I'm hoping the authors can address the analysis Ma et al. included in their Supplementary information section. In that section, a linearized form of the dynamics of three-node networks is analyzed using matrix approaches, similarly to the way it has been done in the current manuscript.

Ref 34 - Robyn P. Araujo and Lance A. Liotta (2018). [And Araujo and Liotta cite, in their Methods, a previous publication by Tang and McMillen (2016), which they describe as an alternative but mathemtically equivalent version of their approach - Zhe F. Tang and David R. McMillen (2016), Design principles for the analysis and construction of robustly homeostatic biological networks. Journal of Theoretical Biology. http://dx.doi.org/10.1016/j.jtbi.2016.06.036]

Ref 34 is addressed briefly in the discussion, but since both of the above papers address the constraints on networks that aim at perfect adaptation/homeostatic disturbance rejection, it would be helpful to have a more detailed discussion of the differences between the current work and the previous reports.

For all of the above-mentioned cases, it would be useful if the authors could provide a more direct discussion of how the current work differentiates itself from that work, presents new or expanded results, and so on. I would recommend that these discussions be included in the manuscript itself, rather than being confined to a response letter.

Reviewer #3: First of all, I would like to sincerely apology to the editors and authors for the delay in my review.

I consider that the article is of interest to the computational biology community. However, the following comments need to be addressed before being accepted for publication.

Major comments:

• 1. Type of networks under study.

There is a major confusion in the article concerning the type of networks under study.

Ma et al, did the study for enzymatic networks (the states of the ODE system are enzyme concentrations). Here, the authors refer all the time to enzymatic networks, but when what they define the system’s under study (starting in line 144), it seems they are talking about GRNs: gene regulatory networks (also denoted as transcriptional networks). Note that the type of kinetics of GRNs (Hill or others), differs from the kinetics of enzymatic networks (Mass action or Michaelis Menten typically). Also, it does not make sense in general to talk about mass conservation in typical GRNs.

The authors need to clarify what type of networks they are dealing with, use the right kinetics, be coherent through the text, and discuss the results with respect to other studies taking this into account. (In the literature, both GRNs and enzymatic networks have been studied in terms of adaptation). The methodology presented iby the authors is independent on the kinetics, and therefore applicable to GRNs, enzymatic, and any other type of biological networks (see comment below).

• 2. Kinetics.

Ma et al, in their work, study enzymatic networks (using a rather particular kinetics), and they suggest in the text that the conclusions for networks involving transcription might differ from those for enzymatic networks. However, the linearization strategy proposed by the authors of this work, justified by the Hartman-Grobman theorem, seems to imply that kinetics are not relevant and the conclusions valid for enzymatic and transcriptional networks. This dichotomy deserves further discussion in the text. Also, if it is valid for general kinetics, it would be better to talk through the text about "biochemical networks" or "biological networks" instead of focusing on a particular class (enzymatic, grn)...

• 3. Mass conservation.

Mass conservation seems to be used through the text without a clear justification, and introduces ambiguity regarding “the order of the system“.

Without mass conservation, as it is the case always in GRNs, N is the number of nodes, the number of states of system, and the number of ODEs. In this case, the meaning of the order of the system=N, is clear.

In line 153, it is stated that “In passing, it may be noted that in the presence of any algebraic constraints on the states (e.g. due to conservation laws), an N-node network corresponds to a reduced order dynamical system” this introduces confusion and might be removed. The order of the system, the number of nodes of a network, and the meaning of N needs to be clarified in the text. If N is used in the text to the number of Nodes, and the number of nodes are the proteins involved, then the number of state variables of the system is <n case="" conservation.="" in="" mass="" of="">

On the other hand, only in cases where X1, X2 and X3 represent different states of the same protein modified through phosphorylation, methylation etc we can have the mass conservation law X1+X2+X3 = constant, otherwise (in case of GRNs for example) it does not make sense. The presence of mass conservation must be "physically" justified through the text.

• 4. Context, originality and generality

The methodology presented (Figure 2) relies on the linearization of the system and then exploiting the theory of linear systems and control theoretic associated concepts (controllability, etc). The authors apply the methodology to study conditions for adaptation.

4.1. The authors justify the use of the linearization using Hartman-Grobman, which they claim it applies in the case of adaptation because, in fact the initial and final states of the process are stable steady states. However, this is not the case for many other functionalities of interest in systems biology, and then the methodology wouldn’t be that general. The conditions under which the methodology applies must then be clearly stated and mentioned from the beginning.

4.2. Other previous studies of adaptation by different approaches (exhaustive exploration, optimization…) do not rely on linearizations. Has linearization been used previously to study adaptation (and/or other motif-functionalities)? A bit of contextualization is missing in this regard, which could clarify better the originality and contribution of the paper. If this is the first it should be somehow emphasized.

4.3. Perfect adaptation is less general than adaptation. The study refers to perfect adaptation and this should be clarified already at the introduction. Also, it is said through the text (line 36) that Ma et al study perfect adaptation but their study deals with adaptation in general.

• 5. Reproducibility. Please provide the full information needed for simulation (not only the equations, also the initial conditions etc) such that the reader can easily reproduce results. It is highly recommended to provide scripts for simulations.

• 6. Models. Where the models in the SI come from? and the particular values of the parameters used? please provide references/context for the models used.

Minor but required

• Citations

Please revise citations along the text, some of them are not correct. Please use the correct and standard form to cite the papers using the first author’s name. For example, use Ma et al (2006) instead of Tang et al (2006). In the same way, it is not “Banga et al (2016) conceived” It is Otero-Muras & Banga (please do not invisibilize the author woman who actually conceived and performed that work). By the way, we further used the MINLP multiobjective optimization methodology to study (robust) adaptation (Otero-Muras and Banga 2019, Processes) in line with previous works by Barnes et al 2011 (PNAS, 108) and Lormeau et al 2017 (IFAC Papers online 50) that also tackled the problem of robust adaptation by different methods.

• Typos. There are several typos along the text (see for example line 620-621). Also, the punctuation marks after the equations are in many cases incorrectly located at the beginning of the next text line.</n>

**Have the authors made all data and (if applicable) computational code underlying the findings in their manuscript fully available?**

Reviewer #1: Yes

Reviewer #2: Yes

Reviewer #3: **No: **to the best of my knowledge they haven't provided the code (not necessary but recommended). Also some data are missing (I have already indicated that in the review).

PLOS authors have the option to publish the peer review history of their article (what does this mean?). If published, this will include your full peer review and any attached files.

Reviewer #1: No

Reviewer #2: No

Reviewer #3: No
---

## [Decision Letter · Decision Letter 1]

16 Dec 2021

Dear Prof Tangirala,

We are pleased to inform you that your manuscript 'Discovering adaptation-capable biological network structures using control-theoretic approaches' has been provisionally accepted for publication in PLOS Computational Biology.

Best regards,

Pedro Mendes, PhD

Associate Editor

PLOS Computational Biology

Jason Haugh

Deputy Editor

PLOS Computational Biology

Reviewer's Responses to Questions

**Comments to the Authors:**

Reviewer #3: The authors have really addressed all the reviewers' concerns in a detailed, exhaustive manner. They have done a great effort in the revision of the manuscript, and made the codes available, which makes the contribution much more useful to the community.

**Have the authors made all data and (if applicable) computational code underlying the findings in their manuscript fully available?**

Reviewer #3: Yes

PLOS authors have the option to publish the peer review history of their article (what does this mean?). If published, this will include your full peer review and any attached files.

Reviewer #3: No

---

## [Editor Report · Acceptance letter]

14 Jan 2022

PCOMPBIOL-D-21-01280R1 

Discovering adaptation-capable biological network structures using control-theoretic approaches

Dear Dr Tangirala,

I am pleased to inform you that your manuscript has been formally accepted for publication in PLOS Computational Biology. Your manuscript is now with our production department and you will be notified of the publication date in due course.

With kind regards,

Katalin Szabo
